# *Aspergillus* Hydrophobins: Physicochemical Properties, Biochemical Properties, and Functions in Solid Polymer Degradation

**DOI:** 10.3390/microorganisms10081498

**Published:** 2022-07-25

**Authors:** Takumi Tanaka, Yuki Terauchi, Akira Yoshimi, Keietsu Abe

**Affiliations:** 1Laboratory of Breeding Engineering for Koji Mold, Department of Biotechnology, Graduate School of Engineering, Osaka University, Suita 565-0871, Japan; takumi_tanaka@bio.eng.osaka-u.ac.jp; 2Terrestrial Microbiology and Systematics, Graduate School of Global Environmental Studies, Kyoto University, Kyoto 606-8502, Japan; terauchi.yuki.76a@st.kyoto-u.ac.jp; 3Laboratory of Environmental Interface Technology of Filamentous Fungi, Kyoto University, Kyoto 606-8502, Japan; yoshimi.akira.8c@kyoto-u.ac.jp; 4New Industry Creation Hatchery Center, Tohoku University, Sendai 980-8579, Japan; 5Laboratory of Applied Microbiology, Department of Microbial Biotechnology, Graduate School of Agricultural Science, Tohoku University, Sendai 980-8572, Japan

**Keywords:** *Aspergillus*, hydrophobin, self-assembly, biopolymer degradation, small secreted protein

## Abstract

Hydrophobins are small amphipathic proteins conserved in filamentous fungi. In this review, the properties and functions of *Aspergillus* hydrophobins are comprehensively discussed on the basis of recent findings. Multiple *Aspergillus* hydrophobins have been identified and categorized in conventional class I and two non-conventional classes. Some *Aspergillus* hydrophobins can be purified in a water phase without organic solvents. Class I hydrophobins of Aspergilli self-assemble to form amphipathic membranes. At the air–liquid interface, RolA of *Aspergillus oryzae* self-assembles via four stages, and its self-assembled films consist of two layers, a rodlet membrane facing air and rod-like structures facing liquid. The self-assembly depends mainly on hydrophobin conformation and solution pH. Cys4–Cys5 and Cys7–Cys8 loops, disulfide bonds, and conserved Cys residues of RodA-like hydrophobins are necessary for self-assembly at the interface and for adsorption to solid surfaces. AfRodA helps *Aspergillus fumigatus* to evade recognition by the host immune system. RodA-like hydrophobins recruit cutinases to promote the hydrolysis of aliphatic polyesters. This mechanism appears to be conserved in *Aspergillus* and other filamentous fungi, and may be beneficial for their growth. Aspergilli produce various small secreted proteins (SSPs) including hydrophobins, hydrophobic surface–binding proteins, and effector proteins. Aspergilli may use a wide variety of SSPs to decompose solid polymers.

## 1. Introduction

Hydrophobins are low-molecular-weight (<20 kDa) amphipathic proteins widely conserved in filamentous fungi. In general, the similarity of amino acid sequences among hydrophobins is very low, but hydrophobins have eight conserved Cys residues, four disulfide bonds, and a specific number of amino acid residues between the Cys residues (C-X_5–7_C-C-X_19–39_-C-X_8–23_-C-X_5_-C-C-X_6–18_-C-X2_–13_ or C-X_9–10_C-C-X_11_-C-X_16_-C-X_8–9_-C-C-X_10_-C-X_6–7_) [1,2]. Hydrophobins have β-barrel structures that are similar to each other [2,3,4,5,6]. Some filamentous fungi such as *Aspergillus*, *Penicillium*, *Trichoderma*, extremophilic species, or mycorrhizal fungi have several to over 10 hydrophobin-encoding genes, whereas many filamentous fungi have only a few such genes [7,8,9,10,11]. The expression profiles of multiple hydrophobin genes depend on the growth stage of filamentous fungi and culture conditions, and cellular localization varies among hydrophobins [12,13,14]. Hydrophobins are secreted by filamentous fungi and self-assemble at solid–liquid or air–liquid interfaces to form amphipathic membranes [15,16,17]. Because formation of such membranes reduces interfacial surface tension, hydrophobins contribute to the formation of aerial hyphae and conidia [18,19,20]. Hydrophobins are specifically accumulated inside aerial hyphae, where they associate with lipid-enriched organelles and may affect the structure and increase longevity of aerial hyphae [7]. Secretion of hydrophobins becomes highest at the sporulation phase, when they form a protective coating of rapidly produced spores [7]. Hydrophobins are involved in the water sensing mechanism of spores and are linked to germination [7]. Hydrophobins coat the surfaces of aerial structures and make these surfaces hydrophobic, which contributes to both conidial dispersal [14] and adsorption of pathogenic filamentous fungi on host insects or plants, whose surfaces are hydrophobic [21,22]. Since hydrophobin-coated hyphae and conidia can escape recognition by the immune systems of animals (e.g., insects, mammals) and plants, hydrophobins are thought to contribute to host infection by pathogenic filamentous fungi [14,23,24]. Hydrophobins attached to solid surfaces are able to recruit and immobilize various proteins such as bovine serum albumin, IgG, avidin, glucose oxidase, horseradish peroxidase, and cutinases [25,26,27,28,29]. Hydrophobins are classified into several classes according to their hydropathy patterns, amino acid sequences, and the solubility of their self-assembled membranes [19,30,31,32,33]. Classification and applications of hydrophobins will be addressed in detail in the next section.

The genus *Aspergillus* belongs to ascomycetes and is a polyphyletic taxon containing many fungi imperfecti [34]. Generally, Aspergilli are highly capable of decomposing solid polymers and have been widely used in the fermentation industry for a long time [35,36]. Currently, Aspergilli are used as the host microorganisms for production of these compounds owing to their high productivity of proteins and primary and secondary metabolites [37,38,39,40]. Aspergilli are used for industrial production of a variety of enzymes, such as amylase, cellulase, glucosidase, hemicellulase, lipase, and phytase from *Aspergillus oryzae* and *Aspergillus niger* [41,42,43,44,45,46,47,48,49,50,51,52], and low-molecular-weight compounds such as itaconic acid from *Aspergillus terreus* [53], citric acid from *A. niger* [54], and kojic acid from *A. oryzae* [55]. Aspergilli can infect animals or plants, and are important in the medical, food, and agricultural and livestock fields [39,56,57,58,59,60]. The whole genomes of major Aspergilli have been sequenced [39,61,62]. Genomic DNA sequences of many Aspergilli are available at the National Center for Biotechnology Information (https://www.ncbi.nlm.nih.gov/, accessed on 8 July 2022) and CAoGD (https://nribf21.nrib.go.jp/CAoGD/, accessed on 8 July 2022). New industrially valuable substances are searched by genome mining, and the mechanisms of pathogenicity are also the focus of ongoing studies [63]. The biological functions of *Aspergillus* hydrophobins have been studied for the last 30 years [24,26,28]. *Aspergillus* hydrophobins form a coating layer on the surface of the cell wall and are involved in infection of animals [24,64,65]. In 2005, Takahashi et al. [26] found that conidial hydrophobin of *A. oryzae* was specifically induced when the fungus was grown on polybutylene succinate *co*-adipate (PBSA) as the sole carbon source. Subsequently, the expression of hydrophobin genes was found to be induced in other filamentous fungi such as *Aspergillus nidulans* [66], *A. niger* [67,68], and *Trichoderma reesei* [69] when these fungi were cultivated on solid polymers of plant origin such as cellulose [67,69] or xylan [69], or on straw [67,68], or steam-exploded sugarcane bagasse [66]. These studies suggest that hydrophobins are also involved in solid polymer degradation by filamentous fungi. Therefore, studying *Aspergillus* hydrophobins will expand our understanding of solid polymer degradation and utilization, and infection of animals [24,64,65] or plants [59,60] by Aspergilli. Because novel properties have been discovered in *Aspergillus* hydrophobins, we expect that other properties and biological functions of hydrophobins will be clarified by studying them in Aspergilli.

The characteristics of hydrophobins from Aspergilli differ from those of other hydrophobins; therefore, studying *Aspergillus* hydrophobins is important for understanding their biological roles. However, no comprehensive analysis of the findings on *Aspergillus* hydrophobins is available. In this review, the physicochemical properties and biochemical and biological functions of hydrophobins produced by Aspergilli are comprehensively discussed on the basis of recent findings.

## 2. Classification and Applications of Hydrophobins

### 2.1. Classification

Hydrophobins are classified mainly into classes I and II [19,30,31,32,33]. Class I includes SC3 from *Schizophyllum commune* [30], EAS from *Neurospora crassa* [16], RodA from *Aspergillus*
*fumigatus* (AfRodA) [65], MPG1 from *Magnaporthe oryzae* [21], DewA from *A. nidulans* [70], and RolA (HypA) from *A. oryzae* [26]. Class I hydrophobins are further subdivided into class IA and class IB according to their origin from ascomycetes or basidiomycetes, respectively [10,71]. Class I hydrophobins form self-assembled structures called “rodlets”, which are similar to β-amyloid fibrils [3,20,72]. Rodlets can be solubilized in trifluoroacetic acid, but are barely soluble in HCl, NaOH, sodium dodecyl sulfate, or ethanol [15,73,74]. In vivo, rodlets can be observed on the surface of aerial structures such as hyphae and conidia [1,13,75] (Figure 1). In vitro, rodlets can form on solid surfaces such as mica and highly oriented pyrolytic graphite (HOPG) [3,76,77,78,79]. Loops Cys3–Cys4, Cys4–Cys5, and Cys7–Cys8 are rich in hydrophobic amino acid residues and do not form specific secondary structures [72]. One or more of these loops may be involved in adsorption to solid surfaces and rodlet formation [2,31,76,78]. Particular hydrophobic amino acid residues in the Cys7–Cys8 loop are essential for both the adsorption to solid surfaces and rodlet formation [31]. It is suggested that hydrophobic residues of the Cys7–Cys8 loop of two hydrophobin molecules form a cross-β core and that continuous elongation of a cross-β sheet results in rodlet formation [76,80] (Figure 2A).

Class II hydrophobins include HFBI and HFBII from *T. reesei* [26,69], HFB4 and HFB7 from *Trichoderma harzianum* [81,82], VDHI from *Verticillium dahliae* [83], and NC2 from *N. crassa* [84]. Class II hydrophobins are found in many ascomycetous filamentous fungi such as *Trichoderma* species [69,81,82,85], *Fusarium* species [83,86], and *Neurospora* species [84], but not in *Aspergillus* species. Class II hydrophobins form self-assembled monolayers that structurally differ from rodlets; these structures can be solubilized in trifluoroacetic acid, HCl, NaOH, sodium dodecyl sulfate, or ethanol [72,87,88]. The self-assembled structures of some class II hydrophobins such as cerato-ulmin can be dissociated through pressure or cooling [88,89]. In vitro, class II hydrophobins form bilayers with hydrophilic domains pointing inward and hydrophobic domains pointing outward, and also form multilayers consisting of stacked bilayers [90]. In general, class II hydrophobin layers show no defined morphology. Atomic force microscopy analysis combined with Monte-Carlo simulation suggests that the self-assembled monolayer of HFBI and HFBII forms a lattice structure in which 3 or 6 molecules are regularly arranged as one unit [91] (Figure 2B,C). Class II hydrophobins have shorter Cys3–Cys4 and Cys7–Cys8 loops than class I hydrophobins, and the hydrophobic region of class II hydrophobins (called “hydrophobic patch”) occupies a smaller part of the molecular surface than that of class I hydrophobins [85]. The Cys3–Cys4 and Cys7–Cys8 loops of class II hydrophobins have random structures; because these loops are short and the proportion of random structures is low, class II hydrophobins do not form rodlets but associate with each other without a large change in their three-dimensional structures to form a regularly aligned self-assembled monolayer [76,78,85]. Molecular dynamics simulation suggests that HFBI adsorbs at the water–oil interface with its hydrophobic patch facing the oil phase without a change in the secondary structure, regardless of the initial orientation [92].

In addition to conventional classes I and II, an intermediate class of hydrophobins, called class III in Aspergilli and pseudo-class I in *Trichoderma* species, has been reported; the pattern of the number of amino acid residues between the Cys residues in different loops in this class is intermediate between those of classes I and II [13,20,93,94]. No consensus has been reached on whether class III and pseudo-class I are the same class. Hydrophobins of an unknown class that cannot be classified in classes I–III or pseudo-class I have also been reported in Aspergilli [12,95]. Compared to the physicochemical properties, biochemical properties, and biological functions of conventional classes I and II, those of non-conventional classes (class III, pseudo-class I, and unknown class) are still poorly understood, except for hydropathy patterns [20,89]. On the basis of the amino acid sequences, class III hydrophobins are predicted to form rodlets that are similar to those of class I hydrophobins [10].

### 2.2. Applications

Various studies on industrial utilization of the unique physicochemical properties of hydrophobins have been underway since ca. 2000. Some examples are listed in Table 1. The amphipathic property of hydrophobins can be applied to the dispersion of colloids. Coating of the hydrophobic surfaces of HOPG, carbon nanotubes, and Teflon particles with hydrophobins makes them hydrophilic [96,97], which improves dispersibility in aqueous solvents [96,97,98,99]. Hydrophobins also enhance the dispersibility of hydrophobic drugs and inhibit drug crystallization [100,101]. Hydrophobins stabilize emulsions and foams, allowing them to be stored for long periods of time [19,102,103,104]. Hydrophobin HFBII associates with enzymes and prevents their unfolding, improving their thermal stability [105,106]. Because hydrophobin self-assembled structures are highly oriented and stable, hydrophobins fused with other polypeptides such as enzymes, domains that can bind other molecules, or peptides targeting specific cells can be used for displaying the polypeptides on solid surfaces with high orientation and density [72,82,107,108,109,110,111,112]. Hydrophobins attached to solid surfaces can interact with some low-molecular-mass proteins or chemicals [25,26,82,113,114]. Coating solid surfaces with some hydrophobins can prevent bacterial adhesion to these surfaces [115]. Since mammalian immune cells seem to hardly recognize *A. fumigatus* conidia coated with AfRodA [24], hydrophobins including AfRodA might be used as coating materials that prevent immune recognition of nano-particles used to deliver drugs to target tissues or organs [24,101,116,117].

## 3. Characteristics of Hydrophobins from Three *Aspergillus* Species

Over 50 potential hydrophobins have been identified in Aspergilli [10]. Even more *Aspergillus* hydrophobins can be predicted from genomic DNA sequences deposited in databases such as those housed at National Center for Biotechnology Information. Among *Aspergillus* hydrophobins, 14 hydrophobins from three species (*A. fumigatus*, *A. nidulans*, and *A. oryzae*) listed in Table 2 have been characterized [12,13,23,26,64,65,70,95]. Hydrophobins of *Aspergillus flavus*, *A. niger*, and *Eurotium rubrum* (synonym: *Aspergillus ruber*) have been identified and studied by proteome analysis [122,123], transcriptome analysis [67,68,124], or observation of conidial surface by scanning electron microscopy [125].

In *A. fumigatus*, seven hydrophobin genes, *AfrodA–G,* have been identified [13,23,65,95]. AfRodA, RodB, RodC, and RodE belong to class I, RodF and RodG belong to class III, and the classification of RodD and RodE is controversial [10,93]. AfRodA is well studied, and only AfRodA is shown to be involved in rodlet formation on the conidial surface (Figure 1) [13,23] and immunological inertia of the conidia [13,24]. The immune system is activated when dectin-1 recognize β-glucan of the fungal cell wall, or dectin-2 or -3 recognize α-mannan of the fungal cell wall [126,127]. AfRodA masks dectin-1 and -2-dependent responses and helps fungal cells avoid immune recognition [128]. Some fungal cell wall proteins such as Ywp1, Erg1, and Lrg1 also mask dectin-dependent responses; however, these proteins are not hydrophobins [129,130]. *AfrodA* is most highly expressed during sporulation, whereas *rodB* is expressed in the biofilm. The transcription of *AfrodA* is controlled by the conidial transcriptional factor BrlA, not by AbaA or WetA [64]. AfRodA, RodB, and RodC are located on the conidial surface [13,23]. Atomic force microscopy investigation of the conidial surface has shown that AfRodA self-assembles into rodlets through bilayers [80]. Within the bilayers, the hydrophobic domains of AfRodA face inwards, making the hydrophobic core. A study of AfRodA structure by NMR spectroscopy and atomic force microscopy has shown that hydrophobic amino acid residues in Cys–Cys loops are important for rodlet formation [31]. Substitution of the conserved Cys residues in AfRodA abolishes the AfRodA secretion to the conidial surface and therefore the rodlet layer [31,131], as reported for MPG1 of *Magnaporthe grisea* [132]. A similar phenomenon has been reported for SC3 of *S. commune*; disruption of disulfide bonds by a reducing agent and free thiol-blocking reagents abolishes rodlet formation by SC3 [133]. The phenylpropanoid isoeugenol inhibits rodlet formation by AfRodA on the conidial surface by decreasing the *AfrodA* transcription level and by interacting with the Cys residues of AfRodA [134].

In *A. nidulans*, six hydrophobin genes, *AnrodA* and *dewA*–*E*, have been identified. The transcription of *AnrodA* is controlled by the conidial transcriptional factor BrlA [135]. The expression of *AnrodA* and *dewA*–*C* has been detected in conidia, but not in vegetative hyphae [12,135]. The expression of *dewD* and *dewE* has been detected both in conidia and hyphae [12]. AnRodA, DewA, and DewB belong to class I, whereas the allocation of DewC–E to a particular class is controversial [12,70]. AnRodA and DewA are well studied, and the structure of DewA has been analyzed by NMR spectroscopy [6]. AnRodA and DewA confer hydrophobicity to the conidial surface, but only AnRodA is involved in rodlet formation on the conidial surface [12,136]. All hydrophobins of *A. nidulans* contribute to colony hydrophobicity [12,136]. DewA–E are involved in cell wall formation. AnRodA, DewA–E are all localized to conidial surface [12]. However, when DewA and DewB are expressed under the control of the AnRodA promoter and the signal peptide from AnRodA is used for secretion, incomplete rodlets are formed on the conidial surface, suggesting that AnRodA can be substituted with neither DewA nor DewB [12]. The coating layer of DewA on glass surfaces, but not those of DewC–E, is stable against ethanol and SDS [136].

Out of several hydrophobins in *A. oryzae*, only RolA (AO090020000588) has been biochemically analyzed. The *rolA* expression patterns and RolA localization are not well characterized. However, RolA is secreted into liquid culture medium when *A. oryzae* is grown in the presence of biodegradable plastic PBSA [26]. It is suggested that the transcription of the *rolA* orthologue in *A. flavus*, a fungus that is considered to have evolved from a common ancestor with *A. oryzae* [137,138], is controlled by the conidial transcription factor BrlA [124]. Two hydrophobic amino acid residues (Leu137, Leu142) in the Cys7–Cys8 loop of RolA are cooperatively involved in RolA adsorption to solid surfaces such as PBSA [139].

## 4. Purification of Hydrophobins and Analysis of Self-Assembly at Interface

Most hydrophobins are purified under denaturing conditions and then refolded because they are highly hydrophobic and aggregate easily [18,23,31,58,140,141,142]. Some hydrophobins are purified by two-phase extraction or reverse-phase high performance liquid chromatography (HPLC) by using their amphiphilic properties [23,78,85,109,121,143]. Only some hydrophobins can be purified in water phase without denaturation, refolding, and high concentrations of organic solvents [17,23,26,28,82,90]. Five *Aspergillus* hydrophobins have been purified (Table 3). RolA, AfRodA, RodB, and AnRodA have been purified in water phase without using organic solvents [26,28,142]. RolA is the only *Aspergillus* hydrophobin that has been purified via a homologous expression system without an affinity tag, denaturing, refolding, and organic solvents [17,26]. RolA secreted into the medium from *A. oryzae rolA*-overexpressing strain has been purified by hydrophobic chromatography, anion exchange chromatography, and cation exchange chromatography, without any affinity tag [17,26]. Recombinant AnRodA has been purified from an *A. oryzae AnrodA*-expressing strain by using the same method as for RolA purification with no affinity tag [28]. AfRodA and RodB secreted into the medium from *Pichia pastoris AfrodA*– or *rodB*–expressing strains have been purified by immobilized metal affinity chromatography with a histidine-tag [142].

Rodlet formation by DewA, RolA, and AfRodA has been analyzed in vitro [6,17,31,144]. At high concentrations, DewA forms dimers but no rodlets [6]. DewA monomers are either conformers A (major type) or conformers B (minor type). Conformers B cannot form dimers but form rodlets more rapidly than conformers A [6]. At the solid–liquid or air–liquid interface, RolA self-assembles to form rodlets. RolA self-assembles at the air–liquid interface to form Langmuir films (membranes) via four stages [17]. RolA Langmuir film undergoes a phase transition from a gas film to a liquid-expanded film, then to a liquid-condensed film, and finally to a self-assembled film. The final self-assembled structures of other hydrophobins, for example, HGFI from *Grifola frondosa* [145] and Vmh2 from *Pleurotus ostreatus* [146,147], have been analyzed, but the process of their self-assembly has not. RolA Langmuir film at the air–liquid interface is structurally different on its hydrophobic and hydrophilic surfaces: a rodlet membrane faces air and rod-like structures face the liquid [17]. At the solid–liquid interface, the self-assembled structure of RolA differs depending on solid surface properties (hydrophobic or charged) and pH conditions, which is attributed to the involvement of charged amino acid residues in the Cys–Cys loops in self-assembly [144]. In addition, the adsorption of RolA depends mainly on the hydrophobic interaction between the solid surface and RolA in the water phase [144]. The interaction between RolA and solid surfaces is also affected by the zeta potential of RolA and the hydrophobicity of its Cys–Cys loops. The structures of assembled RolA differ according to the amount that is adsorbed on solid surfaces [144]. AfRodA self-assembly on the HOPG surface (a solid–liquid interface) has been characterized [31]. Chimeric AfRodA with the central Cys7–Cys8 loop replaced with that of the class II hydrophobin NC2 of *N. crassa* is able to form rodlets. AfRodA mutants with the substitution of one or two hydrophobic amino acid residues in the Cys4–Cys5 loop (I114G, L115G) or Cys7–Cys8 loop (L145G, I146G) also form rodlets. These chimeric AfRodA and AfRodA mutants need longer lag time for self-assembly than does wild-type AfRodA. Peptides corresponding to the Cys4–Cys5 or Cys7–Cys8 loops of AfRodA form fibrils. Therefore, both the Cys4–Cys5 and Cys7–Cys8 loops are involved in rodlet formation by AfRodA [31]. Rodlet formation by hydrophobin EAS requires only the Cys7–Cys8 loop [76]. The involvement of the Cys4–Cys5 loop in hydrophobin self-assembly has been reported so far in AfRodA only [31]. Because Leu145 of AfRodA corresponds to Leu137 of RolA [10], corresponding leucine residues in other RodA-like hydrophobins may be involved in both adsorption to solid surfaces and self-assembly.

## 5. Involvement of Hydrophobins in Solid Polymer Degradation

### 5.1. Hydrophobin–Cutinase Interactions in A. oryzae and A. nidulans

#### 5.1.1. *Aspergillus oryzae*

Direct evidence for hydrophobin involvement in the degradation of solid polymers was first reported in *A. oryzae* in 2005 [26]. This fungus co-expresses RolA and CutL1 when grown on PBSA as a sole carbon source and hydrolyzes the polyester [26,149]. The secreted RolA adsorbs to the PBSA surface [26,139], then it recruits and condenses CutL1 [8,26] (and a CutL1 homologue, CutC [8]), and thus promotes PBSA hydrolysis [8,26] (Figure 3). Cutinases hydrolyze various aliphatic esters such as cutin, PBSA, and triglycerides [149] and are produced by many fungi and bacteria [150,151]. PBSA is structurally similar to cutin, an insoluble wax polyester in the plant protective cuticle [152]. Since both hydrophobin and cutinase are produced by many pathogenic filamentous fungi and promote infection by these fungi [58,150,151,153,154,155,156,157], PBSA degradation via RolA–CutL1 interaction is thought to mimic infection by these fungi [8,26].

Several key characteristics of RolA–CutL1/CutC interaction have been clarified [8,26,158]. 

(1) It is important that RolA adsorbs to the PBSA surface before CutL1 reaches the surface [26]. The PBSA degradation is only slightly accelerated by simultaneous addition of RolA and CutL1 in comparison with the effect of CutL1 alone. RolA secondary structure changes after its adsorption to a solid surface, and this change is necessary for CutL1 recruitment [26].

(2) The adsorbed RolA moves laterally on the PBSA surface but stops moving when CutL1 is added [26] (Figure 3A). Therefore, RolA may act as an anchor or scaffold to tether CutL1. The RolA molecules that do not interact with CutL1 move randomly to expose the PBSA surface to the recruited CutL1.

(3) The recruitment of CutL1 by RolA attached to solid surfaces is driven by ionic interactions between these proteins [8] (Figure 3B). Their interactions are affected by the protonation state of the side chains of amino acid residues in both RolA and CutL1 in a pH-dependent manner. Addition of NaCl prevents these ionic interactions.

(4) Positively charged N-terminal residues His32 and Lys34 of RolA and negatively charged residues Asp30, Glu31, Asp142, and Asp171 on the hydrophilic surface of CutL1 are critically involved in RolA-dependent CutL1 recruitment via ionic interactions [8,158] (Figure 3B). Chemical modification of these charged residues or their substitution with non-charged residues such as serine markedly weaken the RolA–CutL1 interaction. The interactions between the RolA-H32S/K34S mutant and CutL1-E31S/D142S/D171S mutant, and between wild-type RolA and CutL1-D30S/E31S/D142S/D171S are still stronger than the interaction between the wild-type proteins in the presence of NaCl. Therefore, other charged residues (e.g., Lys41, Lys46, and Lys51 of RolA) or complementarity of the three-dimensional structures of RolA and CutL1 may be involved in the interaction.

It cannot be excluded that the properties of cutinases such as substrate specificity and thermal stability may change due to their interaction with RolA, however, this has not been studied yet.

Recently, it has been reported that RolA promotes the degradation of polyethylene terephthalate (PET) by PET-degrading enzyme [148,159], or PETase [160], from the betaproteobacterium *Ideonella sakaiensis*; both PETase and cutinases are alpha/beta-hydrolases. The estimated molecular weight of PETase (27.6 kDa) is about 40% higher than that of CutL1 (19.7 kDa), and the amino acid sequence identity is very low (19.24%; Figure 4A). However, the three-dimensional structures of PETase (Protein databank ID 5XJH; [161]) and CutL1 (Protein databank ID 3GBS; [162]) are similar and some of the negatively charged residues in both proteins are located on the opposite side of the active site (Figure 4B,C). Therefore, the mechanisms of the RolA–PETase and RolA–CutL1 interactions may be similar. Thus, RolA may interact with and recruit various cutinases and cutinase-like enzymes, and thus enhance the hydrolysis of various aliphatic esters by these enzymes.

#### 5.1.2. *Aspergillus nidulans*

The model Aspergilli *A. nidulans* has multiple genes encoding both hydrophobins (Table 2) and cutinases [12,70]. Tanaka et al. reported that hydrophobin AnRodA interacts with cutinases Cut1 and Cut2, promoting PBSA degradation [28]. AnRodA, Cut1, and Cut2 are the orthologues of RolA, CutL1, and CutB of *A. oryzae*, respectively [8,28]. Expression of the *cut1* and *cut2* genes is induced by lipidic carbon sources such as suberin, cutin, or olive oil [163,164,165]. Expression of the *AnrodA* gene is induced by steam-exploded sugarcane bagasse [66], which is composed of cellulose, hemicellulose, lignin, and wax ester [166,167]. In such culture, the activity of extracellular polysaccharide-hydrolyzing enzymes (e.g., cellulases or amylases) and the fungal biomass of the Δ*AnrodA* strain are lower than those of the wild-type strain [66]. Therefore, *A. nidulans* may use AnRodA for the degradation of not only aliphatic esters but also polysaccharides.

AnRodA interacts with Cut1 and Cut2 via ionic interactions in the same way as RolA interacts with CutL1 and CutC [28]. Interestingly, AnRodA also interacts with CutL1 of *A. oryzae* via ionic interactions, although the interaction is much weaker than that between RolA and CutL1 [8]. Positively charged residues in the N-terminus of AnRodA (His23, Lys35, and Lys41) are widely spaced, whereas those of RolA (His32, Lys34, and Lys41) are clustered together in their primary structures [28] (Figure 5). Thus, in the RolA–CutL1/CutC and AnRodA–Cut1/Cut2 interactions [26,28], charged amino acid residues may be in more suitable positions on the surfaces of hydrophobins and cutinases than those in the AnRodA–CutL1 interaction.

### 5.2. Hydrophobin–Cutinase Interactions in Other Fungi

To date, only a few hydrophobin–cutinase interactive combinations have been reported in filamentous fungi, including *A. oryzae* [8,26] and *A. nidulans* [28]. Among other filamentous fungi, the combinations of hydrophobins MPG1 (class I) and MHP1 (class II) and the cutinase Cut2 have been reported in the rice blast fungus *M. oryzae* [29]. However, phylogenetic analysis of hydrophobins and cutinases by Takahashi et al. (Figure 6; [8]) suggests a variety of potential combinations, for example, hydrophobin Pc22g14290 (accession number CAP98717.1)–Cutinase 1 (CAP97019.1) of *Penicillium chrysogenum*, hydrophobin BCDW1_9126 (EMR82223.1)–cutinase BCDW1_3897 (EMR87444.1) of *Botrytis cinerea*, and hydrophobin FVG_03685 (EWG41603.1)–Cutinase 3 (EWG55667.1) of *Fusarium verticillioides*; all accession numbers are from GenBank.

Some class I hydrophobins (Figure 6A) are predicted on the basis of coding sequences only. Many class I hydrophobins, including those of ascomycetes, have multiple positively charged residues in their N-terminal regions upstream of the first Cys residue [8]. Most predicted hydrophobins in the clade containing AnRodA and RolA are from *Aspergillus* and *Penicillium* species [8] and have at least three positively charged N-terminal residues in similar positions (Figure 5). Some class I hydrophobins from other clades also have multiple positively charged N-terminal residues, for instance, Hydpt1 of *Pisolithus tinctorius* (GenBank accession number AAC49307.1; 10 positively charged N-terminal residues), SC6 of *S. commune* (CAA07545.1; 4 positively charged N-terminal residues), and Vmh1 of *P. ostreatus* (CAB41405.1; 4 positively charged N-terminal residues) [168]. Most ascomycetous and basidiomycetous filamentous fungi that harbor hydrophobins (Figure 6A) have several cutinases, including those predicted on the basis of coding sequences. Some filamentous fungi also have acetylxylan esterases of the carbohydrate esterase 5 family (Figure 6B), with amino acid sequences highly similar to those of cutinases. Negatively charged residues corresponding to Glu31, Asp142, and Asp171 of CutL1 are highly conserved in many cutinases of ascomycetes, in some cutinases of basidiomycetes, and in some acetylxylan esterases [8]. The ionic interactions of hydrophobins with cutinases may be common at least in *Aspergillus* and *Penicillium* species, and possibly in many ascomycetes and in some basidiomycetes [8,28,158].

## 6. Low-Molecular-Weight Proteins with Properties Similar to Those of Hydrophobins

Low-molecular-weight proteins (<300 amino acid residues) secreted by filamentous fungi, such as hydrophobins, hydrophobic surface–binding proteins (HsbA [21] and HsbA-like proteins) that do not show a specific pattern of conserved Cys residues characteristic of hydrophobins, and effector proteins, are collectively referred to as small secreted proteins (SSPs) [169,170,171]. In Aspergilli, non-hydrophobin SSPs also attach to solid surfaces and recruit hydrolytic enzymes. HsbA (14.4 kDa) from *A. oryzae* attaches to the PBSA surface in the presence of Ca^2+^ and recruits CutL1 [172]. Similar to the expression of the *rolA* and *cutl1* genes, that of the *hsbA* gene is induced by PBSA [26,149,172,173]. The *hsbA* expression is also induced in solid-state culture with wheat bran [172,174]. Proteins homologous to HsbA and their orthologues are found in *A. niger* and *A. nidulans* [67,68,169]. When these fungi are grown in a medium containing wheat straw, or *A. nidulans* is grown in a medium containing sugarcane bagasse pulp, expression of genes encoding HsbA orthologues is induced [67,68,169]. These observations suggest that HsbA and its orthologues are likely involved in the degradation of solid polymers.

Effector protein is a generic term for multiple protein groups that promote infection by phytopathogenic filamentous fungi and their growth by enabling the fungi to avoid the plant immune response or by damaging plant tissues [175,176,177,178,179,180,181]. Effector proteins have been found in phytopathogenic filamentous fungi at first; however, orthologs of effector proteins also have been found in ectomycorrhizal and saprobic fungi [177,178,182,183]. Hydrophobins and some effector proteins (e.g., the phytotoxin cerato-platanin) have similar physicochemical and biochemical properties such as high hydrophobicity, strong foam formation, self-assembly at the air–liquid interface, and localization on the fungal cell wall [184,185,186,187,188], but have unrelated amino acid sequences [175,186,189,190]. Contrary to the phytotoxicity of effector proteins such as cerato-platanin, hydrophobin toxicity has not been reported. Hydrophobins form hydrophobic protective coating on the surface of the fungal cell wall, and hydrophobins and hydrophobin-coated hyphae and conidia evade recognition by the immune systems of host plants [21,22,23,24]. Although the evasion mechanism has not been well elucidated, the functions of protective coating formation and plant immune response avoidance are common between hydrophobins and some effector proteins [179,180]. The expression of hydrophobin genes is induced in filamentous fungi by solid polymers of plant origin [66,67,68,69]. Therefore, hydrophobins are considered as effector proteins [33,180,191,192,193]. Some other studies suggest that the HsbA-like proteins of *M. oryzae* are also effector proteins because their genes are strongly up-regulated during appressorium development, which is strongly related to host infection [192,194].

In Aspergilli, the number of SSP-encoding genes varies greatly among species [169], and the SSP secretion pattern depends on the plant-derived polymer provided as a carbon source. For example, when the same plant-derived polymer (sugarcane bagasse pulp or wheat bran) is used, one group of HsbA orthologues, which includes HsbA of *A. oryzae*, is barely secreted, whereas another group is secreted on sugarcane bagasse pulp in *A. flavus* and on wheat bran in *Aspergillus clavatus*, *A. niger*, and *A. terreus* [169]. RolA, HsbA, and effector proteins are widely conserved among these Aspergilli [169]. Thus, Aspergilli may decompose plant polymers through the interaction of various SSPs with various polymer-degrading enzymes. The differences in the SSP expression profiles among species suggest that SSP production is optimized in Aspergilli in response to specific solid polymers and environmental conditions, such as salt concentration, pH, and oxidative stress, to decompose the available solid polymers.

## 7. Conclusions

Hydrophobins, low-molecular-weight amphipathic proteins, are widely conserved in filamentous fungi and are localized on the surface of the cell wall. Hydrophobins self-assemble at interfaces and form amphipathic membranes. Class I hydrophobins self-assemble into β-amyloid-like structures called rodlets. Aspergilli have multiple class I hydrophobins. Self-assembly of class I hydrophobins of Aspergilli depends on factors such as hydrophobin conformation, pH of the solution, and the physicochemical properties (e.g., hydrophobicity and functional group) of the solid surface. The Cys4–Cys5 and Cys7–Cys8 loops, four disulfide bonds, and eight conserved Cys residues are all important for the self-assembly of RodA-like class I hydrophobins. The Cys7–Cys8 loop is also important for the adsorption of RodA-like hydrophobins to solid surfaces. Among class I hydrophobins, some *Aspergillus* hydrophobins such as RolA, AfRodA, RodB, and AnRodA can be purified in water phase without using organic solvents. In addition to class I hydrophobins, non-conventional class hydrophobins (class III and unknown class) but no class II hydrophobins have been found in Aspergilli. The physicochemical properties, biochemical properties, and biological functions of non-conventional class hydrophobins are poorly understood, but these hydrophobins may also be important for Aspergilli. Hydrophobins are beneficial for filamentous fungus growth. For example, RolA and RodA-like hydrophobins interact with cutinases to promote the degradation of aliphatic polyesters. This unique mechanism, first discovered in *A. oryzae*, appears to be generally conserved in *Aspergillus* and *Penicillium* species that possess these hydrophobins. It is necessary to further study the mechanism by which self-assembled structures of hydrophobins on solid polymers recruit hydrolytic enzymes and promote hydrolysis of the polymers beneath the hydrophobin self-assembled structures. To the best of our knowledge, the recruitment of enzymes by non-RodA-like hydrophobins of Aspergilli has not been reported but seems plausible because some fungal hydrophobins other than those from Aspergilli also recruit enzymes. The ability of hydrophobins to interact with a variety of enzymes allows the enzymes to be exploited as “functionalized substrates”; other proteins or compounds can be fixed to a solid substrate on which hydrophobins are adsorbed. This concept may be applicable to the fabrication of biosensors, cell culture substrates, and bioreactors for material degradation or conversion (Figure 7). Aspergilli produce various SSPs including hydrophobins, HsbA, HsbA-like proteins, and effector proteins depending on species and culture conditions. Hydrophobins and HsbA interact with polymer-degrading enzymes, recruiting them and thus enhancing solid polymer degradation. Some biochemical properties and biological functions are common between hydrophobins and effector proteins, hydrophobins and HsbA/HsbA-like proteins, and HsbA/HsbA-like proteins and effector proteins. Therefore, Aspergilli may use a wide variety of SSPs to decompose and utilize solid polymers. Further studies from the physicochemical, biochemical, and genetic viewpoints are necessary for understanding the biological roles of *Aspergillus* SSPs.

## Figures and Tables

**Figure 1 microorganisms-10-01498-f001:**
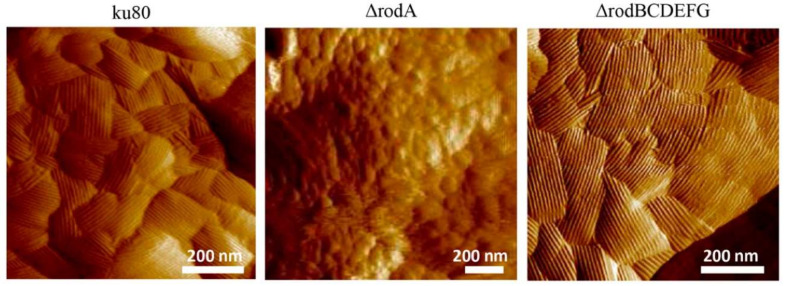
Structure of the conidial cell wall surface of *A. fumigatus* ku80, Δ*rodA*, and Δ*rodBCDEFG*. Atomic force microscopic images show the presence of rodlets on the surface of ku80 and Δ*rodBCDEFG* and their absence on that of Δ*rodA*, in which AfRodA is deleted (Reprinted with permission from [13]. 2017, *Journal of Fungi*).

**Figure 2 microorganisms-10-01498-f002:**
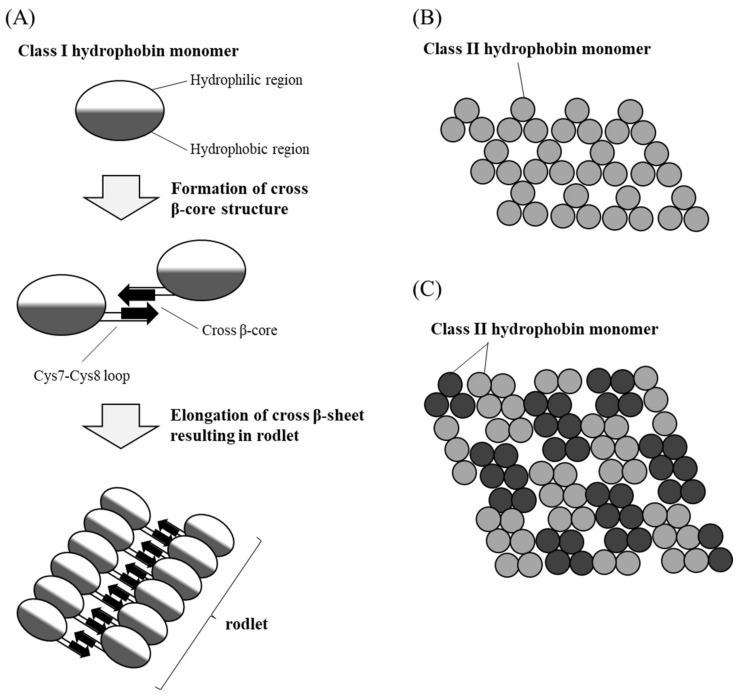
Schematic representation of hydrophobin self-assembly. (**A**) Class I hydrophobin self-assembles into rodlets (based on [76,80]). (**B**,**C**) Self-assembled structure of class II hydrophobin composed of (**B**) 3- or (**C**) 6-molecule units (based on [91]).

**Figure 3 microorganisms-10-01498-f003:**
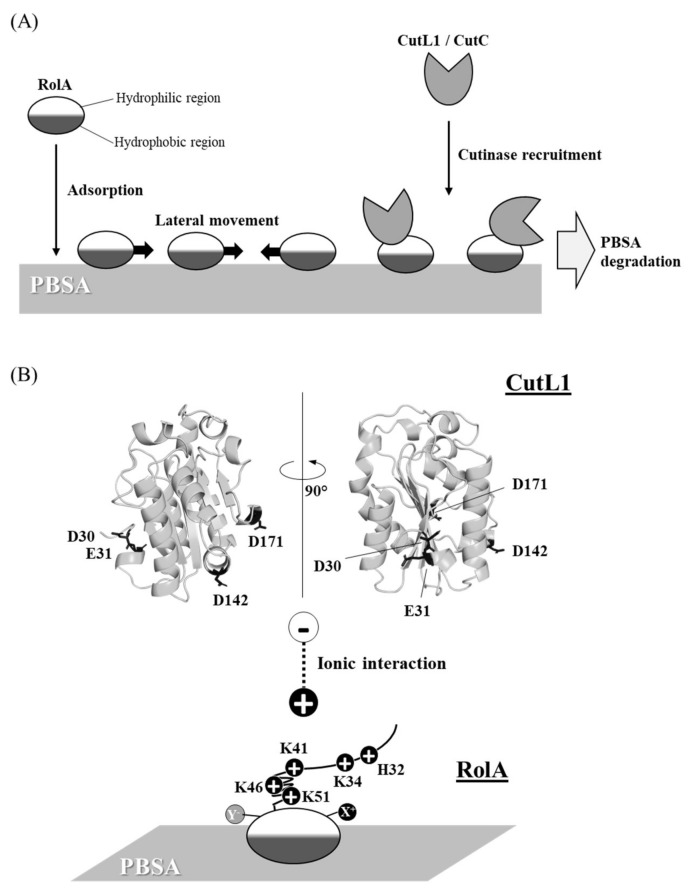
Schematic model of PBSA–RolA–CutL1 interaction. (**A**) Adsorption, lateral mobility, and cutinase recruitment by RolA on the PBSA surface (based on [8]). (**B**) Mechanism of the interaction between RolA and CutL1 (adapted from [8,158]).

**Figure 4 microorganisms-10-01498-f004:**
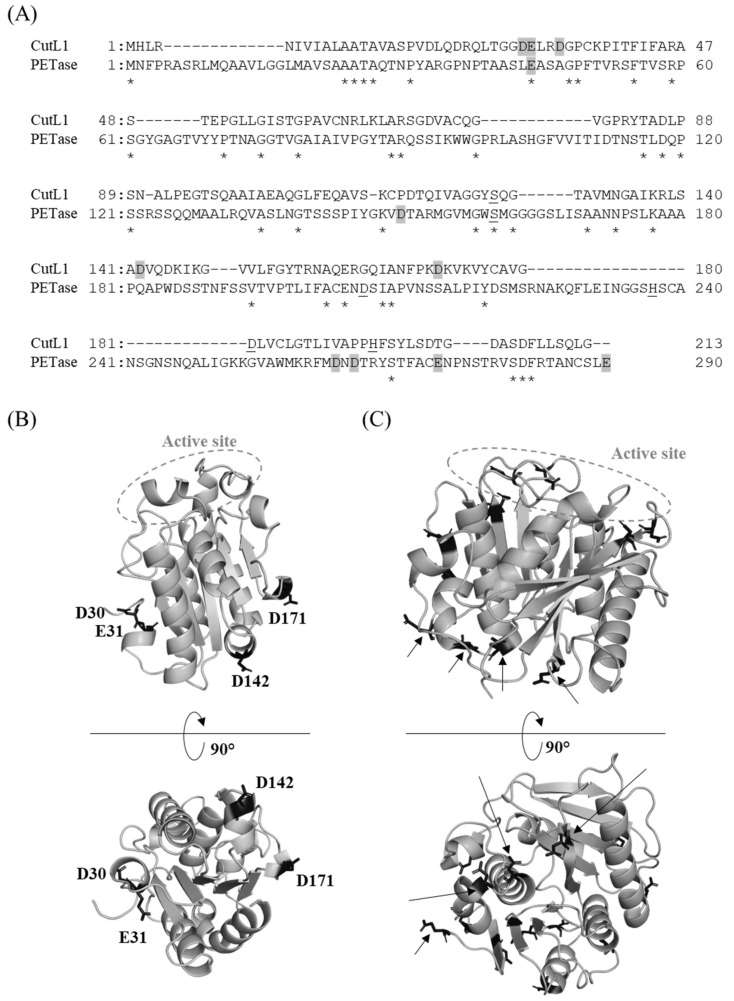
Amino acid sequences and three-dimensional structures of CutL1 and PETase. (**A**) Alignment of the amino acid sequences. Identical residues are indicated by asterisks. Negatively charged residues of CutL1 that are required for the interaction with RolA and negatively charged residues of PETase that are located on the opposite side of the active site are shaded in gray. Catalytic residues are underlined. (**B**) Three-dimensional structure of CutL1. Negatively charged residues that are required for the interaction with RolA are shown as black stick models. (**C**) Three-dimensional structure of PETase. All negatively charged residues are shown as black stick models. Those located on the opposite side of the active site are indicated by arrows.

**Figure 5 microorganisms-10-01498-f005:**
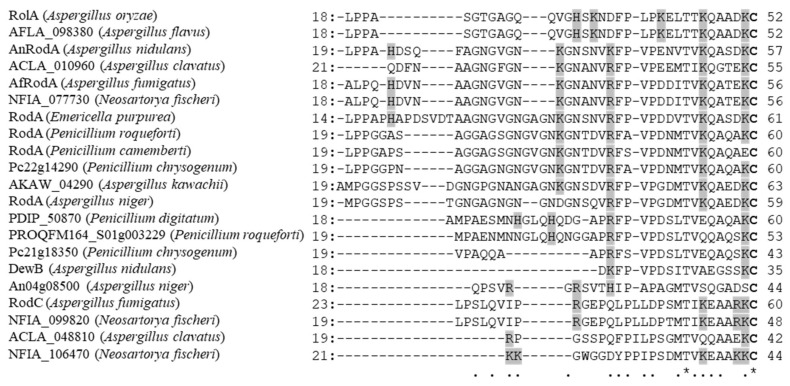
N-terminal regions of hydrophobins in the clade containing AnRodA and RolA (as in [8]). Identical residues are indicated by asterisks, and highly conserved residues are indicated by periods. Positively charged residues (Arg, His, and Lys) are shaded in gray. Cys residues are shown in bold.

**Figure 6 microorganisms-10-01498-f006:**
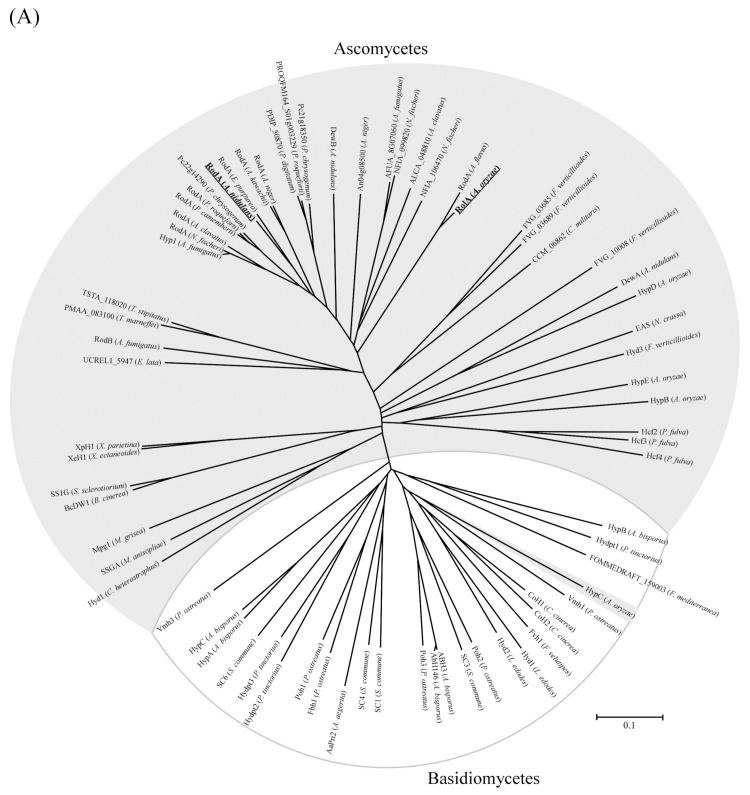
Phylogenetic analysis of (**A**) major class I hydrophobins and (**B**) cutinases and acetylxylan esterases. Acetylxylan esterases are underlined. The cutinases and acetylxylan esterases form the following three groups: (**i**) ascomycetes cutinases, including all Aspergilli cutinases; (**ii**) cutinases from other ascomycetes and basidiomycetes; and (**iii**) acetylxylan esterases and cutinases which show high similarities to acetylxylan esterases. All sequences are from ascomycetes or basidiomycetes (reproduced with permission from [8]. 2015, *Molecular Microbiology*).

**Figure 7 microorganisms-10-01498-f007:**
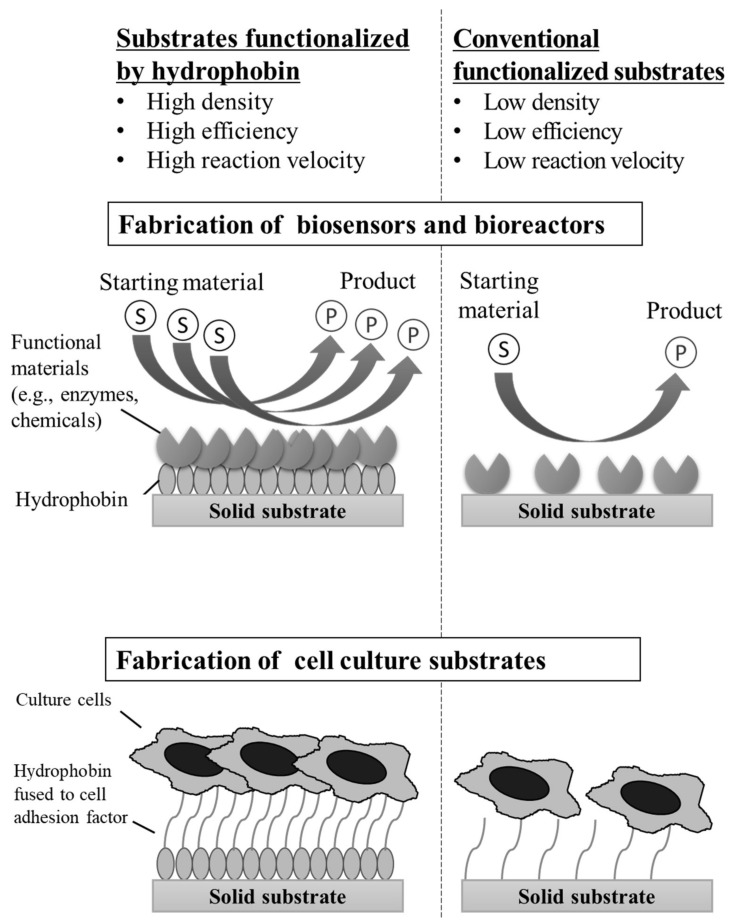
Schematic diagram of hypothetical applications of hydrophobins: fabrication of biosensors, cell culture substrates, and bioreactors for material degradation or conversion. Left part of each panel: solid substrates functionalized by hydrophobins. Right part of each panel: conventional functionalized substrates in which functional materials are immobilized on the solid surface via a chemical reagent.

**Table 1 microorganisms-10-01498-t001:** Expected applications of hydrophobins.

Application	Hydrophobin	Class	Reference
Modification of the wettability of solid surfaces (e.g., Teflon, glass, mica, resin, and stone)	SC3	IB	[118]
	DewA	IA	[119]
	HFBI	II	[119,120]
Enhancement of the dispersibility of hydrophobic particles (e.g., graphene, carbon nano tubes, highly oriented pyrolytic graphite, pigments, and minerals) in aqueous solvents	EAS	IA	[98]
	HGFI	IB	[98]
	HFBI	II	[96,97]
	HFBII	II	[99]
Enhancement of the dispersibility of hydrophobic drugs and inhibition of drug crystallization	HFBI	II	[100,101]
Coating of metal microparticles for medical applications and of drug particles	SC3	IB	[101]
	AfRodA	IA	[24]
	HFBI	II	[116]
	HFBII	II	[117]
Inhibition of bacterial adhesion to solid surfaces	DewA	IA	[115]
Immobilization of functional peptides and proteins (e.g., cell adhesion factor, cellulose-binding module, enzymes, histidine tag, and Protein A) on solid surfaces	DewA	IA	[109,111,113]
	DewB	IA	[111]
	HGFI	IB	[27,112]
	RolA	IA	[26]
	SC3	IB	[25]
	VmhII	IB	[108,110,114]
	HFBI	II	[27,107]
	HFB4	II	[82]
	HFB7	II	[82]
Fusion partner for mass production and efficient purification of recombinant enzymes	HFBI	II	[121]
Enhancement of thermostability of enzymes	HFBI	II	[105,106]
Stabilization of emulsions, bubbles, and foams for long-term storage	SC3	IB	[19]
	HFBI	II	[104]
	HFBII	II	[102,103]

**Table 2 microorganisms-10-01498-t002:** *Aspergillus* hydrophobins that are characterized biochemically.

Organism	Hydrophobin	Number of Cys	Accession Number *	Class	Length, a.a.	Location	Reference
*Aspergillus oryzae*	RolA (HypA)	8	AO090020000588	I	151		[26]
*Aspergillus fumigatus*	AfRodA	8	AFUA_5G09580	I	159	conidia	[65]
	RodB	8	AFUA_1G17250	I	140	conidia	[23]
	RodC	8	AFUA_8G07060	I	155	conidia	[95]
	RodD	8	AFUA_5G01490	unknown	193		[95]
	RodE	8	AFUA_8G05890	I	179		[95]
	RodF	9	AFUA_5G03280	III	190		[13]
	RodG	8	AFUA_2G14661	III	125		[13]
*Aspergillus nidulans*	AnRodA	8	AN8803	I	157	conidia	[64]
	DewA	8	AN8006	I	135	conidia	[70]
	DewB	8	AN1837	I	135	conidia	[12]
	DewC	8	AN6401	unknown	143	conidia	[12]
	DewD	8	AN0940	unknown	101	conidia	[12]
	DewE	8	AN7539	unknown	109	conidia	[12]

* Accession numbers are from the National Center for Biotechnology Information (https://www.ncbi.nlm.nih.gov/, accessed on 23 May 2022).

**Table 3 microorganisms-10-01498-t003:** Procedures for *Aspergillus* hydrophobin purification.

Hydrophobin	Origin	Host for Production	Tag	Purification Procedure	Reference
RolA	*Aspergillus oryzae*	*Aspergillus oryzae*		Hydrophobic chromatography (omittable), anion exchange chromatography, and cation exchange chromatography	[17,26]
		*Escherichia coli*		Denaturation and refolding	[148]
AfRodA	*Aspergillus fumigatus*	*Escherichia col*	His-tag	Immobilized metal affinity chromatography (IMAC)	[142]
		*Pichia pastoris*	His-tag	IMAC	[142]
		*Escherichia coli*	His-tag	IMAC and refolding	[31]
RodB	*Aspergillus fumigatus*	*Aspergillus fumigatus*		Rodlet extraction, denaturation, and reverse-phase HPLC	[23]
		*Pichia pastoris*	His-tag	IMAC	[142]
AnRodA	*Aspergillus nidulans*	*Aspergillus oryzae*		Hydrophobic chromatography, anion exchange chromatography, and cation exchange chromatography	[8]
DewA	*Aspergillus nidulans*	*Trichoderma reesei*	His-tag	Precipitation, denaturation, and refolding	[141]
		*Escherichia coli*	His-tag	IMAC and reverse-phase HPLC	[143]
		*Escherichia coli*	His-tag	Solubilization of inclusion body and refolding	[113]
		*Escherichia coli*	His-tag	Aqueous two-phase separation using isopropyl alcohol	[109]

## Data Availability

Not applicable.

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
