# Peer review of "Aspergillus Hydrophobins: Physicochemical Properties, Biochemical Properties, and Functions in Solid Polymer Degradation"

_microorganisms, 2022, doi:10.3390/microorganisms10081498_

Round 1

Reviewer 1 Report

1.      Don’t use “we describe/state/explain… etc”- it's not scientific. Same for “they” also

2.      Why didn’t you focus on Aspergillus niger as its industrial enzyme producer?

3.      Add more novelty/aim of the work!

4.      Recheck the manuscript regarding low-case and capital letters (ex. Line 189), and italic characters (microorganisms and enzymes).

5.      Manuscript should be revised

6.      Use more newly published citations

Author Response

Thank you for your comments.

We have revised our manuscript.

In “Tanaka_et_al_Main_text_r_highlighted.docx”, all changes in the text were highlighted by yellow, all recent references were highlighted by light blue, and all changes in figures were commented out.

In “Tanaka_et_al_Main_text_r.docx”, all changes from original manuscript were approved.

Below are the point to point answers for your comments.

  1. Don’t use “we describe/state/explain… etc”- it's not scientific. Same for “they” also

→ We have modified the manuscript to avoid the usage you indicated.

  1. Why didn’t you focus on Aspergillus niger as its industrial enzyme producer?

→In the original manuscript (lines 79–85), we cited papers that focus on Aspergillus niger as an industrial enzyme producer, although we did not mention this species explicitly. Therefore, we have modified and expanded the text to explain that Aspergilli, including A. niger, are used as industrial enzyme producers:

Lines 86–94:

Currently, Aspergilli are used as the host microorganisms for production of these compounds owing to their high productivity of proteins and primary and secondary metabolites [37-40]. Aspergilli are used for industrial production of a variety of enzymes, such as amylase, cellulase, glucosidase, hemicellulase, lipase, and phytase from Aspergillus oryzae and Aspergillus niger [41-52], and low-molecular-weight compounds such as itaconic acid from Aspergillus terreus [53], citric acid from A. niger [54], and kojic acid from A. oryzae [55].

  1. Add more novelty/aim of the work!

→ We have modified the “Introduction”. The aim of this review is now stated as follows:

Lines 122–129:

The characteristics of hydrophobins from Aspergilli differ from those of other hydrophobins; therefore, studying Aspergillus hydrophobins is important for understanding their biological roles. However, no comprehensive analysis of the findings on Aspergillus hydrophobins is available. In this review, the physicochemical properties and biochemical and biological functions of hydrophobins produced by Aspergilli are comprehensively discussed on the basis of recent findings.

  1. Recheck the manuscript regarding low-case and capital letters (ex. Line 189), and italic characters (microorganisms and enzymes).

→ We have rechecked the entire manuscript and fixed the text format.

  1. Manuscript should be revised

→ We have revised the manuscript according to the reviewers’ comments.

  1. Use more newly published citations

→ We have added recent references and have highlighted all recent references by light blue in the revised manuscript.

Reviewer 2 Report

In lines 71 and 75 there is a font size change - please inspect it.

Line 91, 93 and 108 change Aspergillus to italic version Aspergillus

line 385 should be written in italic also,

Authors stated that hydrophobins have a toxic effect - I think that there should be a separate paragraph related to that, showing the mechanisms of its toxicity

Is there any possibility that authors would add SEM/TEM images of specific fungi showing their differences?

Does the hydrophobins' interaction with enzymes make them work properly without any changes in their specificity? (like losing the ability to catalyse specific protein reactions? )

Author Response

Thank you for your comments.

We have revised our manuscript.

In “Tanaka_et_al_Main_text_r_highlighted.docx”, all changes in the text were highlighted by yellow, all recent references were highlighted by light blue, and all changes in figures were commented out.

In “Tanaka_et_al_Main_text_r.docx”, all changes from original manuscript were approved.

Below are the point to point answers for your comments.

  • In lines 71 and 75 there is a font size change - please inspect it.

→ These size changes likely occurred when the submission system automatically converted the manuscript format. We have checked the entire manuscript and corrected font size.

  • Line 91, 93 and 108 change Aspergillus to italic version Aspergillus

→ We have italicized “Aspergillus” in all instances of “Aspergillus hydrophobins” and “Aspergillus SSPs”.

  • line 385 should be written in italic also,

→ Corrected (lines 391 and 474 in the revised manuscript).

  • Authors stated that hydrophobins have a toxic effect - I think that there should be a separate paragraph related to that, showing the mechanisms of its toxicity

→ Although toxic and non-toxic effector proteins are known, hydrophobin toxicity has not been reported. Biochemical properties and functions of SSPs including effector proteins are not the main focus of this manuscript, but similarities and differences between hydrophobins and effector proteins may be important for understanding the biochemical properties and functions of hydrophobins. We have separated the second paragraph of section 6 into two paragraphs and modified the text. Similarities and differences between hydrophobins and effector proteins, and the reason for the inclusion of hydrophobins in effector proteins are described in the first of the split paragraphs as follows:

Lines 572–597:

Effector protein is a generic term for multiple protein groups that promote infection by phytopathogenic filamentous fungi and their growth by enabling the fungi to avoid the plant immune response or by damaging plant tissues [175-181]. Effector proteins have been found in phytopathogenic filamentous fungi at first, however, orthologs of effector proteins also have been found in ectomycorrhizal and saprobic fungi [177,178,182,183]. Hydrophobins and some effector proteins (e.g., the phytotoxin cerato-platanin) have similar physicochemical and biochemical properties such as high hydrophobicity, strong foam formation, self-assembly at the air–liquid interface, and localization on the fungal cell wall [184-188], but have unrelated amino acid sequences [175,186,189,190]. Contrary to the phytotoxicity of effector proteins such as cerato-platanin, hydrophobin toxicity has not been reported. Hydrophobins form hydrophobic protective coating on the surface of the fungal cell wall, and hydrophobins and hydrophobin-coated hyphae and conidia evade recognition by the immune systems of host plants [21-24]. Although the evasion mechanism has not been well elucidated, the functions of protective coating formation and plant immune response avoidance are common between hydrophobins and some effector proteins [179,180]. The expression of hydrophobin genes is induced in filamentous fungi by solid polymers of plant origin [66-69]. Therefore, hydrophobins are considered as effector proteins [33,180,191-193]. Some other studies suggest that the HsbA-like proteins of M. oryzae are also effector proteins because their genes are strongly up-regulated during appressorium development, which is strongly related to host infection [192,194].

  • Is there any possibility that authors would add SEM/TEM images of specific fungi showing their differences?

→ We have added a new figure (Figure 1 in the revised version) that shows atomic force microscopic images of rodlets on the conidial surface (Lines 142–144, 157–160 and Figure 1).

  • Does the hydrophobins' interaction with enzymes make them work properly without any changes in their specificity? (like losing the ability to catalyse specific protein reactions?)

→ We cannot exclude that the properties of enzymes such as substrate specificity may change due to their interaction with hydrophobins, but as far as we know, this has not been studied. We have added a sentence in the manuscript as follows:

Lines 443–445:

It cannot be excluded that the properties of cutinases such as substrate specificity and thermal stability may change due to their interaction with RolA, however, this has not been studied yet.

Round 2

Reviewer 1 Report

There is no comment.